# Transcriptome Dynamics during Spike Differentiation of Wheat Reveal Amazing Changes in Cell Wall Metabolic Regulators

**DOI:** 10.3390/ijms241411666

**Published:** 2023-07-19

**Authors:** Junjie Han, Yichen Liu, Yiting Shen, Donghai Zhang, Weihua Li

**Affiliations:** College of Agriculture, The Key Laboratory of Oasis Eco-Agriculture, Xinjiang Production and Construction Group, Shihezi University, Shihezi 832000, China; hanjunjie1208@sina.com (J.H.); lyc162710@163.com (Y.L.); 15662712664@163.com (Y.S.); xjzhangdh@126.com (D.Z.)

**Keywords:** wheat inflorescence, cell wall, gene expression, xyloglucan endotrans glucose/hydroxylase (XTH), transgenosis

## Abstract

Coordinated cell proliferation and differentiation result in the complex structure of the inflorescence in wheat. It exhibits unique differentiation patterns and structural changes at different stages, which have attracted the attention of botanists studying the dynamic regulation of its genes. Our research aims to understand the molecular mechanisms underlying the regulation of spike development genes at different growth stages. We conducted RNA-Seq and qRT-PCR evaluations on spikes at three stages. Our findings revealed that genes associated with the cell wall and carbohydrate metabolism showed high expression levels between any two stages throughout the entire process, suggesting their regulatory role in early spike development. Furthermore, through transgenic experiments, we elucidated the role of the cell wall regulator gene in spike development regulation. These research results contribute to identifying essential genes associated with the morphology and development of wheat spike tissue.

## 1. Introduction

Wheat is a crucial food crop, providing nutrition and energy for one-third of the global population [1]. With the challenge of rapid population growth, increasing wheat yield has become a primary goal for breeders worldwide. The inflorescence structure of wheat plays a vital role in seed yield, encompassing panicle development, spikelet formation, and floret development [2]. Therefore, conducting comprehensive research on the regulatory mechanisms governing wheat spike development is essential for improving yield and quality.

Understanding the development and regulation of inflorescences is of great importance to plant biologists and crop breeders, as it is closely linked to the reproductive processes and grain production in flowering plants. Wheat’s compound spike consists of multiple sessile spikelets interlacing along the spike axis. Each axis node gives rise to several spikelet meristems (SMs). These SMs typically produce two to four fertile, small flowers that develop into seeds [3]. Unlike other crops, wheat lacks branches, and the central flower axis directly produces spikelets. The number of SMs per flower axis influences the number of small flowers produced, and occasional variations in spikelet excess may occur [4]. While critical genes involved in inflorescence initiation and development are conserved in higher plants [5,6], there is still a need for a comprehensive understanding of the diversity and behavior of inflorescences.

Previous studies in rice and maize have identified multiple genes associated with spike development using functional genomics, bioinformatics, and genetic resources [7,8]. However, our understanding of wheat spike development’s molecular mechanisms remains relatively limited. Genes and quantitative trait loci (QTLs) related to spike development are dispersed across wheat’s 21 chromosomes [9]. For example, the *Q* gene on chromosome 5A is closely linked to spike compactness and affects spike length, plant height, and spike emergence time [10]. The *TaSnRK2* gene, found on chromosomes 4A/4B/4D, encodes a protein kinase that influences spike length and thousand-grain weight [11,12]. Mutations in the *ARGONAUTE1d* gene in durum wheat result in shorter spikes and fewer grains per spike [13]. The compact (*C*) gene on chromosome 2D plays a critical role in spike compactness, grain size, shape, and grain number per spike [14]. Additionally, the *Photoperiod1* (*Ppd1*) gene on chromosome 2D inhibits paired spikelet formation by regulating the expression of the *FT* gene [15]. The polyploid nature of wheat, its large genome size (approximately 17 Gb), and low transformation efficiency have posed challenges in understanding the complex genetic network governing spike development. Nevertheless, these studies have provided insights into the molecular mechanisms involved in spike development.

The differentiation of wheat spikes is a complex biological process influenced by multiple genetic and physiological factors. Transcriptome dynamics, which refer to changes in gene expression patterns over time, play a crucial role in wheat spike differentiation and yield formation [16]. By studying transcriptome dynamics, we can unveil the temporal and spatial distribution patterns of gene expression in wheat panicles at different developmental stages. This helps us understand the molecular mechanisms underlying wheat panicle differentiation. Recent advancements in high-throughput sequencing and metabolomics technologies have allowed for a comprehensive investigation of dynamic changes in the wheat transcriptome during spike differentiation. Extensive genome sequencing efforts have been undertaken to construct reference sequences for wheat [17,18], culminating in the latest version of the wheat reference genome, RefSeq v2.1, available on the IWGSC website (http://www.wheatgenome.org/ (accessed on 5 January 2023)). It provides a unique opportunity to systematically unravel the gene expression regulatory network and the metabolic regulatory network during wheat spike differentiation. We can enhance our understanding of wheat spikes’ formation and development mechanisms by studying transcriptome dynamics, providing a valuable scientific basis for improving wheat yield and quality. Significant differences exist in the composition, developmental process, and function of cell walls (CWs).

In addition to cellulose, wood glucan (a hemicellulose polysaccharide), pectin, lignin, and other significant components [19], additional components are involved. CWs have multiple functions, including providing mechanical support, regulating intercellular flow, and serving as a barrier to environmental stress. During the development of inflorescences, the formation, differentiation, and shape of CWs are usually influenced by physical, chemical, and biological reactions. These developmental processes are regulated by a variety of function-specific enzymes, including glycosyltransferases (such as cellulose-like synthetase CslC) and xyloglucan endoglycosyltransferases/hydrolases (XTH) [20]. Meanwhile, cell wall-modifying enzymes such as β-1,4-glucanase and xylanase are also involved in reconstructing and modifying cell walls. By regulating the composition and structure of cell walls, these enzymes affect the morphological establishment and functional functioning of flower organs [21]. During inflorescence development, changes in the morphology and composition of cell walls are crucial for forming floral organs. By increasing or decreasing the activity of specific enzymes, the synthesis and degradation of cell walls are balanced, thereby affecting the plasticity and stability of cell walls. This regulatory mechanism enables the cell wall to adapt to different environmental conditions and biological processes.

This study aims to elucidate the dynamic changes in gene expression during the development of wheat spikes. We assessed temporal variations in transcriptional abundance throughout spike development. We aimed to unravel the molecular mechanisms underlying wheat apical meristem development. We specifically focused on describing stage-specific differences in cell wall development and metabolism within the transcriptome. Furthermore, we investigated the subcellular localization of *TraesCS7A02G426700*. Notably, we generated transgenic wheat lines that overexpressed the *TraesCS7A02G426700* gene and performed evaluations of spike traits in these transgenic lines.

## 2. Results

### 2.1. Analysis of Transcriptome Expression Profile and Differentially Expressed Genes during Wheat Spike Development

After conducting RNA-Seq analysis, we detected differentially expressed genes (DEGs) at different stages of wheat spike development: pistil and stamen primordium differentiation stage (S1), anther separation stage (S2), and tetrad formation stage (S3). After removing low-quality and adapter sequences, we obtained 49,552,576, 49,254,077, and 46,632,094 clean reads, respectively. We conducted a comparison between S2 and S1, and S3 and S2, and then identified many DEGs. A total of 12,688 DEGs were identified from these two libraries using Cufflinks software (2.2.1), considering the absolute value of fold change (log2FC > 1 or log2FC < −1) and statistical significance (*p*-value < 0.05) for each gene. Through volcano plots, we visualized the transcriptome dynamics of S2 vs. S1 (Figure 1A) and S3 vs. S2 (Figure 1B). After removing duplicate genes, there were 9493 remaining DEGs, with 7312 genes upregulated (Figure 1C) and 2181 genes downregulated (Figure 1D).

Genome-wide transcriptome analysis demonstrates widespread gene expression during spike development. We employed a strict 5% false discovery rate (FDR) screening to compare the number of upregulated and downregulated DEGs between consecutive time points. Significance analysis (Figure 1) reveals that, in the comparison of S2 vs. S1, we detected 4049 significant DEGs, with 2952 showing upregulation and the rest demonstrating downregulation. However, in comparing S3 vs. S2, we identified 8639 significant DEGs, with 6452 upregulated and 2187 downregulated. The number of DEGs in S3 vs. S2 is twice that of S2 vs. S1, indicating that the latter period may hold more importance in biosynthesis and energy metabolism.

### 2.2. STC and GO Annotation Analysis of DEGs

We observed eight distinct gene expression profiles after conducting an STC analysis of the DEGs (Figure 2). Out of these eight profiles, we identified four significant expression profiles and ranked them based on their respective *p*-values (Figure 2A). Further statistical analysis of these four significant profiles revealed that upregulated and downregulated genes accounted for 81.45% of the total DEGs (see Appendix A for details). Moreover, among the remaining four insignificant expression profiles, we observed that the expression level of the Profile 1 gene decreased during the S1–S2 transition and stabilized during the S2–S3 transition. Conversely, the expression level of the Profile 6 gene increased during the S1–S2 transition. Notably, Profile 2 and Profile 5 exhibited contrasting expression patterns (Figure 2A). Regarding the four significant expression profiles, Profile 4 and Profile 0 contained the highest number of DEGs, with 4167 and 811 genes, respectively (Figure 2B, Appendix A).

We conducted GO annotation and enrichment analysis to understand the gene profiles of four statistically significant differential expression patterns involved in spike development. Appendix A presents the results of GO annotations. Based on the annotated *p*-values, the profiles belong to upregulated Profiles 4 and 7 and downregulated Profiles 0 and 3. Figure 3 shows the distribution in the enrichment analysis (*p* < 0.05). Profile 4’s most crucial enrichment term is GO: 0003824 (catalytic activity). Additionally, GO: 0016787 (hydroxylase activity), GO: 0016491 (oxidoreductase activity), GO: 0005975 (carbon metabolic process), GO: 0071554 (cell wall organization or biogenesis), and GO: 0042546 (cell wall biogenesis) are also significantly enriched (Figure 3A). In Profile 0, the first three significantly overexpressed terms are GO: 0043226 (organelle), GO: 0043229 (internal organelle), and GO: 0003676 (nucleic acid binding) (Figure 3B). In Profile 7, the most significant enrichment term is GO: 0016787 (hydrolase activity), while GO: 0005975 (carbohydrate metabolic process) and GO: 0015979 (photosynthesis) are also significantly enriched (Figure 3C). In Profile 3, the first three enriched terms are GO: 0035639 (purine ribonucleotide triphosphate binding), GO: 0032559 (adenyl ribonucleotide binding), and GO: 0005524 (ATP binding) (Figure 3D). Functional annotations related to carbohydrate and cell wall metabolism were frequently detected in all four profiles, indicating that both functions play essential roles in spike development.

In addition, we performed GO analysis on all DEGs, and the results are listed in Appendix A and displayed in Figure 3E (*p* < 0.05). We observed significant enrichment of items related to DNA replication (GO: 00044815, GO: 0032993, GO: 0004427, GO: 0005694, GO: 0000785, GO: 0006323, GO: 0031497, GO: 0006333, and GO: 0065004). This enrichment suggests that these genes are involved in cell division and tissue development by regulating the cell cycle and gene expression during panicle development. Furthermore, hydrolase activity (GO: 0016798, GO: 0016798, GO: 004553, GO: 0016818, and GO: 0016817), carbohydrate metabolism (GO: 0005975, GO: 0044262, GO: 008643, GO: 0034637, and GO: 0016051), and cell wall metabolism (GO: 0071554, GO: 0042546, GO: 0070592, GO: 0044038, and GO: 0009834) were also significantly enriched. These findings further support the importance of carbohydrate and cell wall metabolism in panicle development and indicate that DNA replication and hydrolase activity also play vital roles in this process.

### 2.3. MapMan Analysis Uncovered the Involvement of Active Cell Walls and Carbohydrate Metabolism during Spike Development

We analyzed the transcriptome data using MapMan and presented the results through the following steps: Firstly, we defined the functional categories. Secondly, significantly overexpressed functional groups were identified and displayed across different stages of spike development (refer to Figure 4 and Appendix A). During the early stage of spike differentiation, carbohydrate metabolism becomes active, providing energy for spike differentiation and elongation. Furthermore, the widespread expression of genes associated with cell wall metabolism indicates their crucial involvement in cell wall synthesis, remodeling, intercellular communication, cell division, and expansion.

During spike development, the activation of glycolysis, the tricarboxylic acid (TCA) cycle, and mitochondrial electron transport are essential energy sources [22,23]. Our gene expression data demonstrate that most sucrose degradation and glycolysis genes are up or downregulated during the S1–S2 stage (Figure 4). Key enzymes, including sucrose invertase, hexokinase, phosphofructokinase (PPFK), and pyruvate kinase (PK), exhibit notable changes in expression. For instance, from the S1 to S3 stages, the expression of genes *TraesCS7D02G008700* (encoding sucrose invertase) and *TraesCS1D02G123700* (encoding hexokinase) decreased by 23 and 2 times, respectively. In contrast, the expression of genes *TraesCS3D02G109600* (related to PPFK) and *TraesCS4A02G129900* (involved in PK) was strongly activated and upregulated during the S1–S2 and S2–S3 stages. In addition, sucrose synthase (SUSY) plays a crucial role in starch synthesis by converting sucrose into fructose and glucose. Following that, UDPG pyrophosphorylase (UGPase) facilitates the conversion of fructose and glucose into uridine diphosphate glucose (UDPG). UDPG acts as a glucose donor in synthesizing glycosides, oligosaccharides, polysaccharides, and related molecules. Our analysis revealed significant upregulation of genes encoding SUSY (*TraesCS7D02G036600*, *TraesCS4A02G446700*, and *TraesCS7A02G040900*) and UGPase (*TraesCS7B02G319900* and *TraesCS7A02G419300*) during spike development. Previous studies have also demonstrated that the interaction between UGPase and ADPG pyrophosphorylase (AGPase), another key enzyme in starch synthesis, generates ADPG for starch synthesis [24,25]. Our data indicate dynamic changes in the expression of multiple genes related to AGPase, such as *TraesCS5D02G182600*, *TraesCS5D02G484500*, and *TraesCS7B02G183300*. The upregulation of these genes contributes to the gradual accumulation of carbohydrates.

Our findings reveal a significant expression of genes associated with cell wall metabolism during spike development. The synthesis of plant cell walls involves two consecutive steps that play a fundamental role in shaping and strengthening cells. We observed changes in the expression of cell wall-related genes throughout spike development, with significant differences at different growth stages. During the transition phase from S1–S2, we observed notable upregulation of specific genes, including glycosyltransferases (*TraesCS4B02G323100*, *TraesCS4D02G320100*, and *TraesCS3B02G474200*), endotransglucosylase/hydrolases (*TraesCS7A02G426700* and *TraesCS7A02G427000*), endoglucanases (*TraesCS4A02G248200*, *TraesCS4D02G065600*, and *TraesCS4B02G066700*), and cellulose synthases (*TraesCS2A02G102600*, *TraesCS6B02G104600*, and *TraesCS5D02G517200*). These genes are crucial for forming primary cell walls in rice [26,27,28]. The high expression levels of these genes suggest that forming primary cell walls involves synergistic degradation, biosynthesis, and remodeling assembly of xylan, glucan, and xylan.

In the S2–S3 stage, we observed significant upregulation of fatty acyl CoA reductases (*TraesCS4B02G283200*, *TraesCS4A02G304400*, and *TraesCS4D02G282000*), ATP-binding cassette transporters (*TraesCS1D02G053800* and *TraesCS1A02G051800*), phosphatidyl diacylglycerol acyltransferase 1 (*TraesCS5A02G207500* and *TraesCS5D02G213600*) involved in an acyl-coenzyme-independent pathway to generate triacylglycerol, and BAHD acyltransferase 1 (*TraesCS1B02G354000*, *TraesCS1A02G341300*, and *TraesCS1D02G343400*) involved in secondary wax/keratin biosynthesis [29]. These genes are associated with fatty acid synthesis and the assembly of secondary cell walls, including suberin and keratin synthesis from fatty acids and lignin synthesis from Phenylalanine [30,31]. These findings indicate that primary and secondary cell wall assembly occurs at different stages of spike development.

During spike development, we used MapMan cell response visualization tools to analyze the reactions associated with the cell wall and carbohydrate metabolism. We extracted genes involved in these two processes separately and conducted an enrichment analysis to determine if these transcriptional changes were specific to developmental stages (Appendix A). Most transcripts associated with the cell wall and glucose metabolism exhibited upregulation during the S2–S3 stage (Figure 5A). During the S1–S2 stage, there was enrichment of significant DEGs (*p* < 0.05) in critical metabolic pathways, including “Cutin, suberin, and wax biosynthesis”, “Fatty acid degradation”, and “Carbohydrate digestion and absorption” (Figure 5B). In the S2–S3 stage, a greater enrichment of significant DEGs related to these two metabolic pathways was observed (Figure 5C). It is noteworthy that the 4-Coumarate:CoA ligase (such as *TraesCS6B02G294100*, *TraesCS2B02G291100*, and *TraesCS4B02G269200*) is involved in the synthesis of ferulic acid in the phenylpropane metabolic pathway and displays upregulation (Appendix A). Previous studies have highlighted the crucial role of ferulic acid oligomers in building the extensive molecular network of the cell wall through cross-linking with hemicellulose chains and hemicellulose–lignin complexes in the cell walls of Poaceae plants [32,33].

### 2.4. Validation of Differentially Expressed Genes by qRT-PCR

To validate the accuracy of RNA-Seq analysis results, we employed quantitative real-time polymerase chain reaction (qRT-PCR) to confirm the expression of 15 DEGs (Figure 6). These DEGs consist of two oxidation-related genes (*SOD* and *POD*), eight genes related to cell wall metabolism (*CSI1*, *Ces*, *XTH*, *MUR3*, *XI*, *GLC1*, *EG,* and *BGL*), one gene associated with fatty acid metabolism (*FAR*), and four genes involved in glycometabolism (*SS*, *HK*, *PK*, and *SST*). The expression patterns of these 15 DEGs exhibited consistent trends between RNA-Seq and qRT-PCR results, indicating the accuracy of the transcriptome analysis. The details of the gene-specific primers used in this study can be found in Appendix A.

### 2.5. Experimental Verification of Candidate Spike Regulator Gene

The transcriptome analysis results have provided abundant gene resources for screening genes related to wheat spike development, particularly those associated with cell wall development. These findings hold significant implications for enhancing wheat grain production capacity. To validate this hypothesis, we selected an essential gene involved in cell wall generation, xyloglucan endotransglucosylase/hydrolase (*TraesCS7A02G426700*), for experimental validation. Notably, we observed the expression of *TraesCS7A02G426700* throughout all three stages of spike development, with continuous upregulation in the S1 to S3 stages (Appendix A). We proceeded to overexpress this gene in the locally cultivated wheat “Xinchun 11” in Xinjiang, generating over 20 independent transgenic lines. We conducted phenotype analysis on two of these lines.

The *TraesCS7A02G426700* gene encodes a protein sequence that exhibits high homology to *AtXTH25*. In Arabidopsis, researchers have shown that this protein influences inflorescence and seed development by regulating cell wall metabolism [34]. In our study, we discovered that the transgenic plant line carrying *TraesCS7A02G426700* exhibited longer spikes (Figure 7A), indicating the potential role of this gene in regulating spike elongation. To investigate the cellular localization and potential functions of the *TraesCS7A02G426700* protein, we fused its coding region with the N-terminal of the GFP gene. We expressed it under the control of the CaMV35S promoter. Microscopic observation revealed that the TraesCS7A02G426700-GFP fusion protein was present on the cell wall (Figure 7B), suggesting that TraesCS7A02G426700 is a cell wall protein. Furthermore, we observed a significant prolongation of the development time for the S1, S2, and S3 stages in the *TraesCS7A02G426700* overexpressing strain (Figure 7C), accompanied by a significant increase in the number of spikelets and spike length (Figure 7D,E). In conclusion, we hypothesize that the *TraesCS7A02G426700* gene plays a crucial regulatory role in spike development by modulating cell wall degradation processes.

## 3. Discussion

The development of wheat spikes directly affects yield, as a healthy and well-developed wheat spike typically produces more grains. Previous studies have conducted large-scale transcriptome analyses on young inflorescences of maize [35], rice [36], and barley [37]. This identifies modules that regulate spike development. However, we still need to improve our understanding of genes related to wheat spike development [38,39]. Therefore, this study aims to explore the molecular pathways that regulate the development of wheat panicles through dynamic transcriptome analysis of young wheat panicles at the stages of pistil and stamen primordium differentiation, anther separation stages, and tetrad formation. We will compare differentially expressed genes (DEGs) and identify highly expressed genes during spike development, focusing on DEGs involved in cell wall metabolism. We aim to provide new insights into the gene regulation of inflorescence development in wheat by analyzing the transcriptome data from critical stages in wheat inflorescence development.

The complexity of ear development determines its production potential. Therefore, regulating the complexity of ear development is an essential strategy for improving yield potential. We found that the increased gene types strongly depend on ear maturity by comparing differentially expressed genes and identifying highly expressed genes during wheat ear development (Figure 3 and Figure 4). Upon investigating the core set of differentially expressed genes, we discovered that these highly expressed genes mainly involve catalytic enzyme activity, hydrolase activity, carbohydrate metabolism, cell wall tissue, and biosynthesis (see Figure 3). It is worth noting that during the three stages of spike development, the number of upregulated differential genes (7312) was significantly higher than the number of downregulated differential genes (2181). It may reflect a general enhancement of the apical meristem before the complete termination of spike formation [40]. Most DEGs exhibited upregulation during the S1–S2 stage in the transcriptome data. This stage, characterized by the emergence and differentiation of stamen meristems, is crucial in wheat inflorescence development. Previous studies have highlighted the significance of this stage, as it ultimately influences the nutrition of tiny flowers and the grain count per spike [41]. Our research has identified several protein genes associated with cell wall metabolism during this stage, including 4-collate: coenzyme A (4CL). 4CL acts as a critical enzyme in the lignin biosynthesis pathway, providing precursors for lignin synthesis by catalyzing the esterification of phenylpropanoic acid and coenzyme A. We observed an upregulation of 4CL gene expression, thus promoting lignin biosynthesis. Furthermore, the transcriptome data revealed heightened activity of genes related to other cell wall components such as suberin and keratin (derived from fatty acids) and those involved in the lignin pathway (derived from Phenylalanine). Previous research has shown that flower organs’ formation and flowering processes involve the regulation of various enzymes, including lignin enzymes. These enzymes’ activity and gene expression levels exhibit changes at different stages of flower development. Upregulation of cellulase and lignin enzyme expression promotes cell wall degradation, which is beneficial for elongating flower organs [42]. This study presents many related gene expression patterns. Based on these findings, we propose a regulatory process that appears to dominate spike development as the stages progress (Figure 8). It suggests that the synthesis and degradation of cell walls are extensively involved in wheat spike development, governing their differentiation and elongation.

Cell walls are the main components of plant morphology and contribute to the biomechanics of organs through their rigid or viscoelastic properties. The cell wall contributes to the shape and function of plant organs by defining the organ-strengthening effect of vascular and fibrous cells with thickened cell walls on the surface of individual cells and inside. Cell wall synthesis is a necessary process in the development of panicles or inflorescences. Cellulose synthase, hemicellulose synthase, and lignin synthase regulate cell wall synthesis. Research has shown that these enzyme genes’ expression levels and activities exhibit spatiotemporal specificity at different stages of spike or inflorescence development [43]. The growth and development of flowers require regulating cell size and shape to control changes in cell wall plasticity. Studies on *Mirabilis jalapa* [44], *Rosa* sp. [45,46], and *Sandersonia aurantiaca* [47] have shown that the development and opening of flowers involve cell wall metabolism. Another study suggests that inflorescence growth mainly depends on cell expansion, and the cell wall is the main limiting factor for cell expansion [48]. Therefore, genes related to cell wall metabolism may contribute to cell wall modifications related to inflorescence development. This result is supported by the high expression of cell wall-related protein genes detected in gladiolus’s stamen, petal, and tea [49].

It is particularly noteworthy that xyloglucan endotransglucosylase/hydrolase (XTH) participates in the dissociation of cell walls [50,51]. Our results support the critical role of xyloglucan as a primary cell wall component in the transition of wall tissue during spike development in monocotyledons, such as wheat. Interestingly, we observed high expression of some genes involved in two different pathways of plant cell wall biosynthesis, and the functions of these enzymes have been confirmed in previous reports. These genes are directly involved in the biosynthesis of cork and keratin precursors in multiple plant organs, including panicles. They include *CHS* [52], *CCR* [53], *4CL* [54], *GPAT* [55], *ADH* [56], *KCS* [57], and keratin/cork transport processes, as well as ATP binding cassette (ABC) transport proteins located in the plasma membrane [58]. In addition, during the S2–S3 stage, we also observed significant upregulation of genes encoding acyltransferases (Ats) and peroxidases (POD), which play essential roles in the synthesis of cork, keratin, and lignin (Appendix A). Recent transcriptome data have also shown that gene expression in cell wall metabolism exhibits stage dependence during spike differentiation [16]. In this study, we observed the upregulation of genes related to the biosynthesis and remodeling of xyloglucan during the S2–S3 conversion. Since xyloglucan endotransglucosylase (XET) is directly involved in the dissociation of cell walls, our research results further support the essential role of xyloglucan as the main component of cell walls during the transformation of spike differentiation in monocotyledons, such as wheat. Based on our research findings, a possible inference is that the active synthesis process of cell walls predominates in the later stages of spike development.

Our results indicate that the overexpression of the *TraesCS7A02G426700* gene, which encodes the XTH protein, can influence spike development in wheat. Wheat lines overexpressing *TraesCS7A02G426700* exhibit long spikes. To evaluate whether changes in cell wall structure cause differences in spike traits, we compared the microstructure of wild-type and transgenic plants. The transgenic plants displayed irregular and distorted shapes in the spike axis, stem under the spike, small flowers, and ovary cells, resulting in a more expansive gap space and less compact cells (Appendix A). Previous studies have reported the impact of *XTH* genes on tissue development and morphogenesis. For instance, overexpression of the *XTH9* gene promotes the elongation of reproductive meristem and inflorescence cells in Arabidopsis [59]. *XTH1*, another studied gene, enhances cell wall flexibility by relaxing the cell wall and degrading the cellulose wood glucan matrix, facilitating rapid expansion and maturation of fruit [60]. *XTH1* also affects the cell wall structure of transgenic Arabidopsis by altering the cellulose structure by encoding a corn cell wall binding protein [61]. The *TraesCS7A02G426700* gene shares a high sequence similarity with these genes, suggesting its potential role in cell wall-related processes. Changes in the cell wall structure, possibly influenced by the overexpression of *TraesCS7A02G426700*, may lead to increased interstitial space and less compact cells. These changes can impact the reorganization and reconstruction of cell walls, potentially affecting cell division and expansion. Consequently, they can influence the duration of the three stages (Figure 7C) and the number of spikelets (Figure 7D).

In conclusion, our study employed transcriptome data analysis and transgenic verification to identify genes associated with spike development, offering potential avenues for enhancing spike traits in wheat. By effectively regulating the expression of genes involved in cell wall metabolism, it is possible to augment both the length and number of spikelets, thereby maximizing the spikelet production potential.

## 4. Materials and Methods

### 4.1. Plant Materials and Growth Conditions

On the experimental farm of Shihezi University, we cultivated the spring wheat “Xinchun 11”, which is primarily grown in Xinjiang. The experiment comprised three biological replicates, with each replicate covering an area of 10 square meters. The cultivation management followed field production conditions. To conduct transcriptome analysis, we randomly selected 50 spikelets from the main branches at different developmental stages, namely, the pistil and stamen primordium differentiation stage (S1), the anther separation stage (S2), and the tetrad formation stage (S3). We manually removed the leaves surrounding the young spikes to collect the samples. We carefully excised the reproductive tissue without stems using a sharp blade under a stereomicroscope (S8 APO, Leica Microsystems) to identify the developmental stage [62,63]. Simultaneously, we collected approximately 30 spikes at each stage for RNA extraction. The collected samples were immediately submerged in liquid nitrogen and stored at −80 °C for future use.

### 4.2. Gene Function Annotation and Enrichment Analysis

Appendix A provides detailed information on RNA extraction, RNA-Seq library preparation and sequencing, and bioinformatics analysis. We obtained the reference genome sequences and gene annotations from the Ensemble plant database (https://ftp.ensemblgenomes.ebi.ac.uk/pub/plants/release-56/ (accessed on 5 January 2023)). The genome sequences underwent quality filtering to remove low-quality bases, and short reading segments were filtered out. Subsequently, we mapped the RNA sequencing reads to the annotated gene assembly (Appendix A). To identify DEGs between the control and experimental groups, we employed the F-test of the Random Variance Model (RVM) in a small sample scenario to ensure higher degrees of freedom. We conducted a significance analysis and applied a False Discovery Rate (FDR) threshold to screen for statistically significant DEGs [64]. We used the Cufflinks program v2.2.1 (http://cole-trapnell-lab.github.io/cufflinks/ (accessed on 5 January 2023)) to determine the expression levels of all DEGs, with specific details provided in the reference [65]. The expression level of each gene was normalized using Fragments Per Kilobase of exon per Million mapped reads (FPKM). Furthermore, we utilized the DAVID bioinformatics resources (v2022q4) (https://david.ncifcrf.gov/home.jsp (accessed on 5 January 2023)) to classify the identified differentially expressed genes into Gene Ontology (GO) categories.

### 4.3. Series Test of Cluster (STC) Analysis of DEGs

By applying the ANOVA corrected by the RVM, we employed the series test of the cluster (STC) to identify DEGs. Our analysis revealed a unique set of expression pattern trends based on the distinct changes in signal density among genes under different circumstances. We transformed the original expression values into log2 ratios. We further defined specific patterns using a clustering strategy with short-term sequence gene expression data. The expression model characterizes the observed or predicted number of genes involved in each expression trend within the model. To assess the significance of the identified patterns, we employed Fisher’s exact test and multiple comparison tests to determine if their occurrence exceeded the expected probability [66].

### 4.4. MapMan Analysis

For the MapMan analysis, we created the input file by calculating the ratio of the natural logarithm of three control samples to the average detection value of the processed samples. Genes with two missing values out of three replicates were deemed unexpressed under the corresponding experimental conditions. We employed MapMan version 3.6.0RC1 [67] for the final analysis, which automatically applied the Wilcoxon rank sum test.

### 4.5. RNA Extraction and Real-Time Quantitative PCR

RNA extraction from 100–200 mg of frozen tissue was conducted following the guidelines provided by the manufacturer, utilizing the TransZol Up Plus RNA Kit (Lot# Q41020, TransGen, Beijing, China). We assessed the quality of the extracted RNA using the Nanodrop 8000 (Thermo Fisher Scientific Inc., Logan, UT, USA). At the same time, the quantity was determined using the Agilent Bioanalyzer 2100 (Agilent Technologies Inc., Santa Clara, CA, USA). We employed the EasyScript One-Step gDNA Removal and cDNA Synthesis Super Mix (Lot# P20708, TransGen, China) for reverse transcription. The Actin gene (GenBank accession number: KC775782.1) was the reference gene. The control group comprised plants of the same age. Following the manufacturer’s protocol, we conducted three independent biological replicates using the PerfectStartTM Green qPCR SuperMix (TransGen Biotech, Beijing, China). We performed the qRT-PCR analysis using the ABI QuantStudio™ 6 system (ABI, Carlsbad, CA, USA). The 2^−ΔΔCT^ method was employed to normalize the relative gene expression levels. Appendix A lists the primers used for quantitative PCR.

### 4.6. Construction of pCAMBIA1301-TraesCS7A02G426700 Vector for Agrobacterium-Mediated Wheat Transformation

We designed specific primers for the full-length coding sequence (CDS) of the *TraesCS7A02G426700* gene. The obtained PCR product, 870 bp in length, was cloned into the pMD18-T vector. Subsequently, we inserted the *TraesCS7A02G426700* gene into the pAHC25 vector containing the Ubi promoter. Finally, we transferred the entire gene construct into the pCAMBIA1301 vector. After sequencing to confirm its integrity, we used the resulting pCAMBIA1301-TraesCS7A02G426700 vector for the Agrobacterium-mediated transformation of wheat. To verify the presence of the target gene and selective marker genes in the transgenic plants, we selected a minimum of 10 individual plants from the T0 generation for GUS staining and resistance analysis using hygromycin (150 μg/mL, Vetec, Sigma, China). Similar analyses were performed on the T1 and T2 generations to support subsequent research.

### 4.7. Subcellular Localization of The TraesCS7A02G426700

The PCR product of the *TraesCS7A02G426700* gene was cloned into the pBWA (V) HS-gfp vector (Biorun Biosciences Co., Ltd., Wuhan, China) to create a fusion vector called CaMV35S-TraesCS7A02G426700-GFP. After sequencing, we transferred the fusion and control vector (pBWA (V) HS-gfp) into Agrobacterium tumefaciens strain GV3101. We transformed tobacco leaves using the method Yang et al. [68] described. We cultured the transformed leaves on MS medium for 48 h and performed live cell imaging using inverted confocal microscopy (Zeiss LSM 780, Jena, Germany).

### 4.8. Microscopic Observation of WT and Transgenic Plants

During the heading stage of WT and T2 transgenic wheat plants, we collected samples from specific regions: the middle of the spike axis (0.5 cm), the stem 1–2 cm below the spike, small flowers, and the ovary before pollination. The samples were promptly fixed in FAA solution (5% formaldehyde, 6% acetic acid, and 45% ethanol) under vacuum conditions for 1 h. Afterward, a sequential process was conducted, including dehydration in 70%, 85%, 95%, and 100% ethanol (1 h each), followed by a transition from 100% ethanol to 100% xylene. Subsequently, the samples were permeated and embedded in paraffin. Using a slicing machine (JY202A, Beijing, China), 6 μm thickness sections were cut and mounted onto microscope slides. Safranine staining was applied, and the samples were examined using an optical microscope and imaged using a confocal laser scanning microscope (Zeiss LSM 800 with Airyscan, Jena, Germany).

### 4.9. Data Analysis

We employed the statistical software SPSS V20 (SPSS, Inc., Chicago, IL, USA) to perform all analyses. An ANOVA was employed to investigate gene expression differences, followed by Fisher’s LSD tests. Additionally, differences between means were detected using Duncan’s multiple range tests with a significance level set at *p* < 0.05.

## Figures and Tables

**Figure 1 ijms-24-11666-f001:**
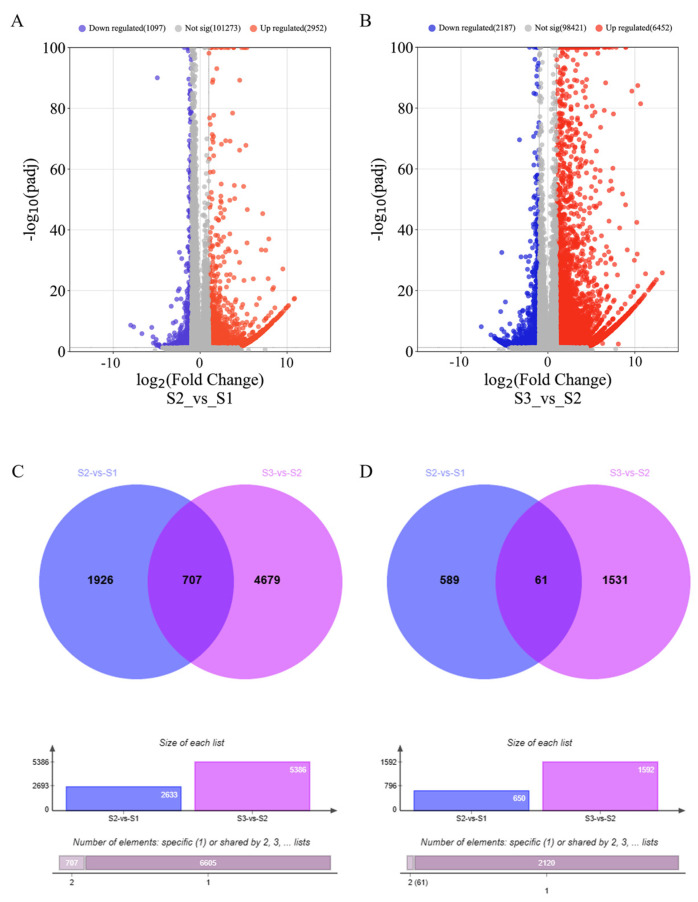
The volcano plot displays the correlation between expressed genes. (**A**) S2 vs. S1 transition. (**B**) S3 vs. S2. Red spots, log2 fold change > 1 and *p*-value < 0.05; blue spots, log2 fold change < −1 and *p*-value < 0.05. The Venn plot depicts the overlapping differentially expressed genes (DEGs) that are upregulated (**C**) and downregulated (**D**) at different stages of wheat spike development. The accompanying bar chart displays the total number of DEGs across different comparative groups.

**Figure 2 ijms-24-11666-f002:**
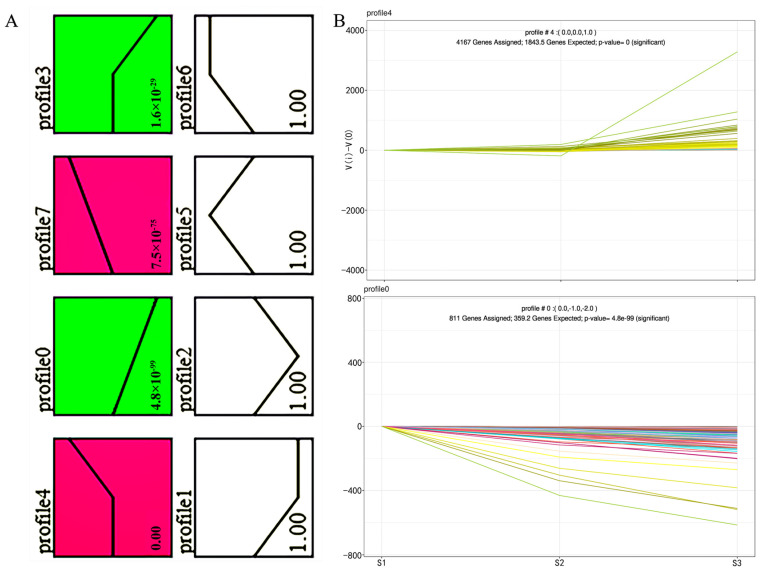
Gene expression patterns were analyzed using model maps, and two significant patterns (Profiles 4 and 0) were displayed. (**A**) Each box in the figure represents a model expression profile. The upper number in each profile box indicates the model number, while the lower number represents the *p*-value associated with different gene expression patterns. Importantly, all four gene expression patterns exhibited significant *p*-values (*p* < 0.05) and were labeled with the same color when they shared the same expression pattern. (**B**) Profile 4 demonstrates an increasing expression level during spike development, whereas Profile 0 shows a decreasing expression level during the same period. The horizontal axis represents the developmental stage, while the vertical axis represents the time series of gene expression levels after log_2_-normalization transformation. The numbers near the profile represent the utilized trend models, while each line represents a gene in the sample.

**Figure 3 ijms-24-11666-f003:**
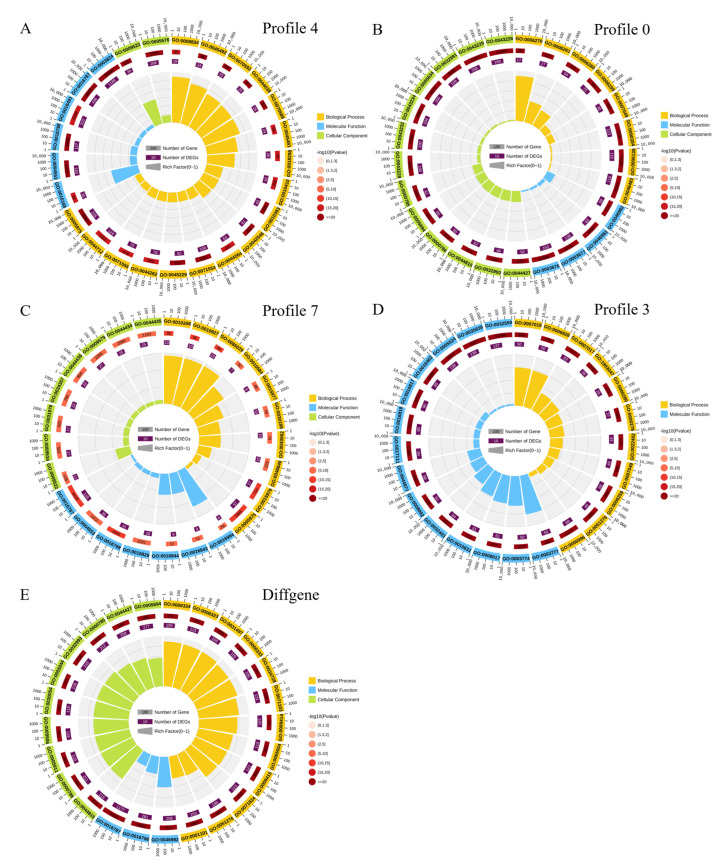
GO annotation analysis of the four significant expression profiles. Panels (**A**–**D**) represent the GO analysis results for the four significant expression profiles, while Panel (**E**) represents the GO analysis results for all DEGs. In the figure, the outer circle represents the GO classification, where the size of each circle indicates the scale of the number of genes, and different colors represent different classifications. The second circle displays the number of genes in each GO classification within the background gene set and the corresponding *p*-value. The length of the bar represents the number of genes, and the color becomes redder as the *p*-value decreases. The third circle represents the number of differentially expressed genes. The inner circle indicates the RichFactor values for each category, with each small grid on the gridline representing 0.1.

**Figure 4 ijms-24-11666-f004:**
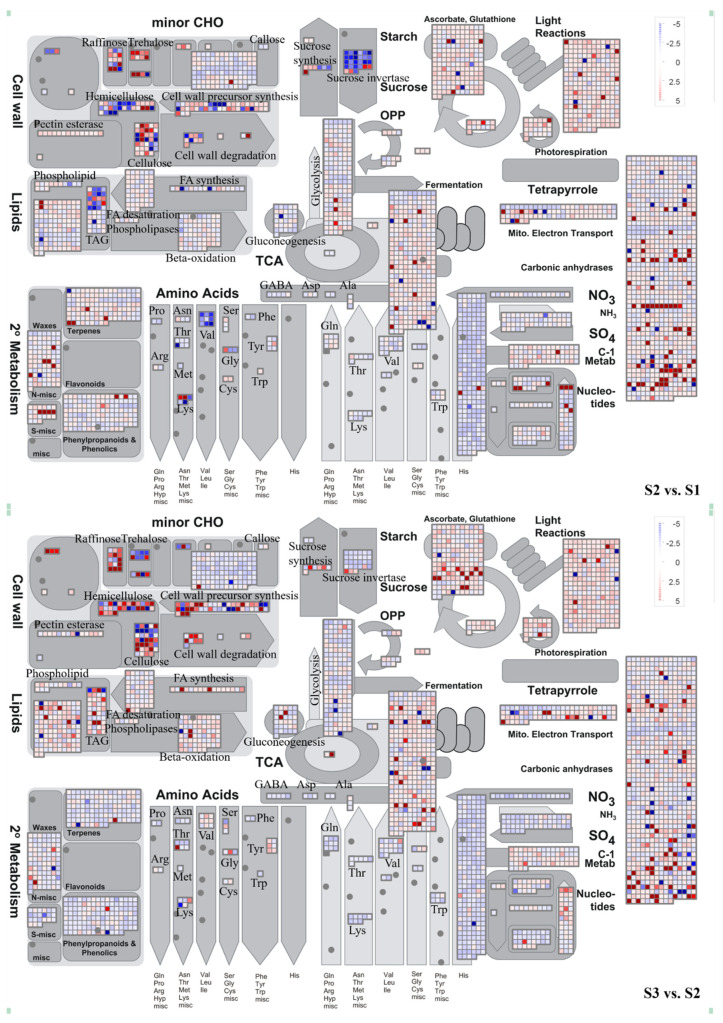
MapMan metabolism overview maps show differences in transcript levels (S2 vs. S1 and S3 vs. S2) during spike development. log2 ratios for average transcript abundance were based on three replicates of RNA-seq. The resulting file was loaded into the MapMan Image Annotator module to generate the metabolism overview map. Blue represents downregulated transcripts on the logarithmic color scale, and red represents upregulated transcripts.

**Figure 5 ijms-24-11666-f005:**
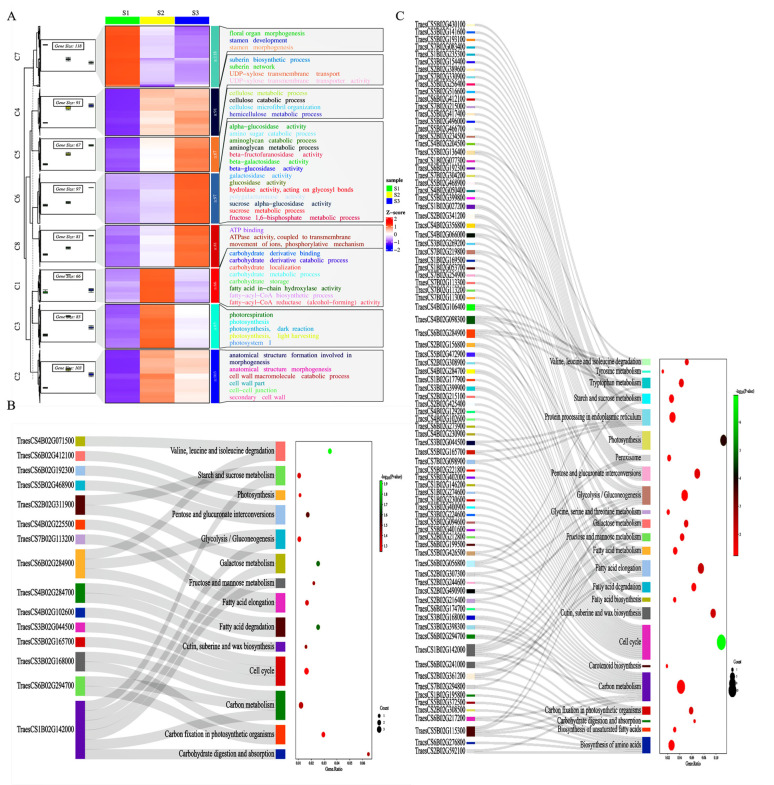
GO classification and pathway enrichment of genes related to the cell wall and carbohydrate metabolism. (**A**) Gene expression heatmap and classification. The left side shows that the transcriptome data are divided into different clusters, and the heat map is generated according to log_2_FPKM. The right side displays different GO classifications. In S1–S2 (**B**) and S2–S3 (**C**), pathways corresponding to significant DEGs were demonstrated. The horizontal direction represents the proportion of genes, while the vertical direction represents specific pathways. The color of the dot represents the *p*-value, and the size of the dot represents the number of genes.

**Figure 6 ijms-24-11666-f006:**
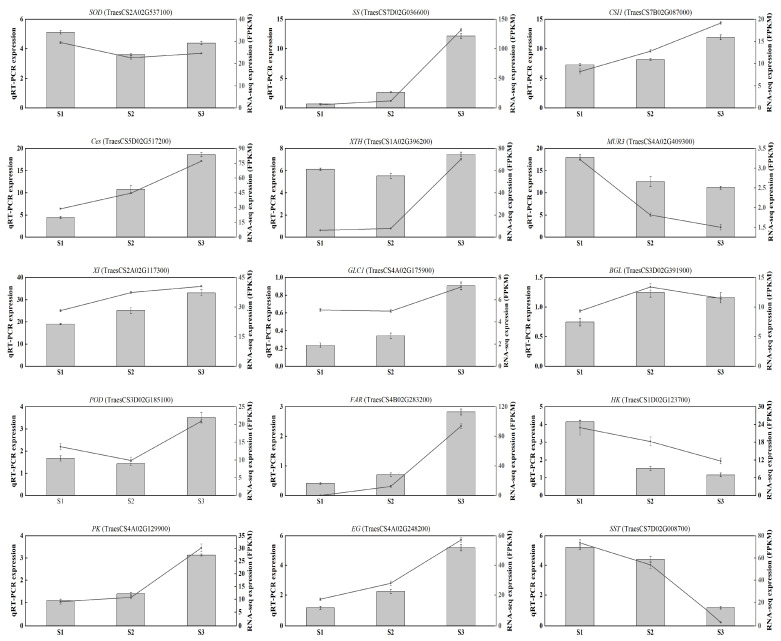
Perform qRT-PCR using 15 randomly selected DEGs. The bar chart and line chart represent qRT-PCR and RNA-seq data, respectively. The data are expressed as the mean ± standard error (SE).

**Figure 7 ijms-24-11666-f007:**
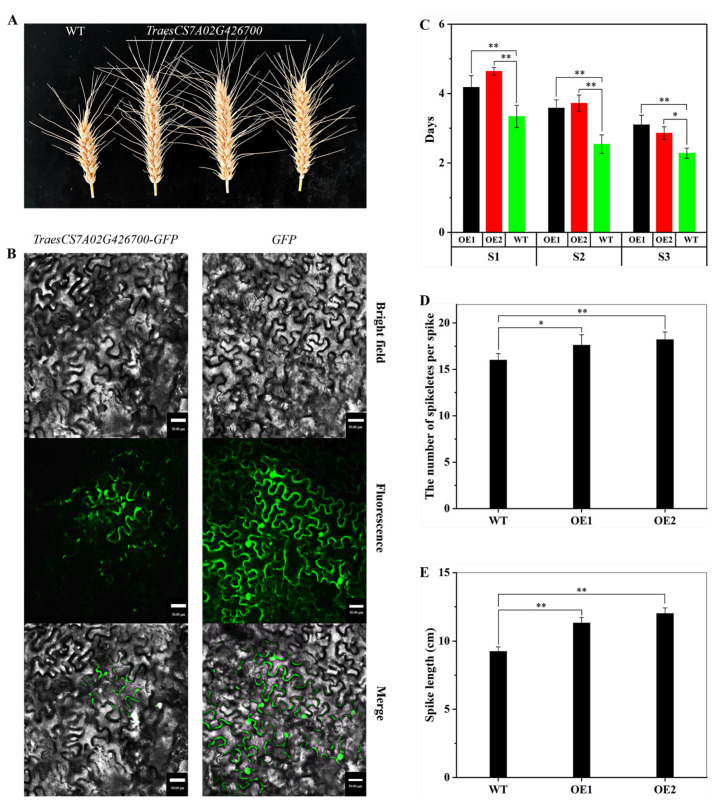
Functional verification and subcellular localization of wheat transgenic plants. (**A**) Spike morphology of WT and T2 transgenic plants. (**B**) Subcellular localization of *TraesCS7A02G426700* in N. *benthamiana* leaves. The control vector (GFP) and fusion construct (*TraesCS7A02G426700-GFP*) were separately transiently expressed in five-week-old N. *benthamiana* leaves by agroinfiltration, and all images were collected under the Zeiss confocal microscope after agroinfiltration for 48 h. The bar is 50 μm. (**C**) Comparing the developmental duration between the wild type (WT) and transgenic lines (OE1 and OE2) at each stage, as well as examining the number of spikelets per spike (**D**) and spike length (**E**) in T2 transgenic plants. Data are the mean ± SD of 10 plants for each line. Student’s *t*-test, * *p* < 0.05, ** *p* < 0.01.

**Figure 8 ijms-24-11666-f008:**
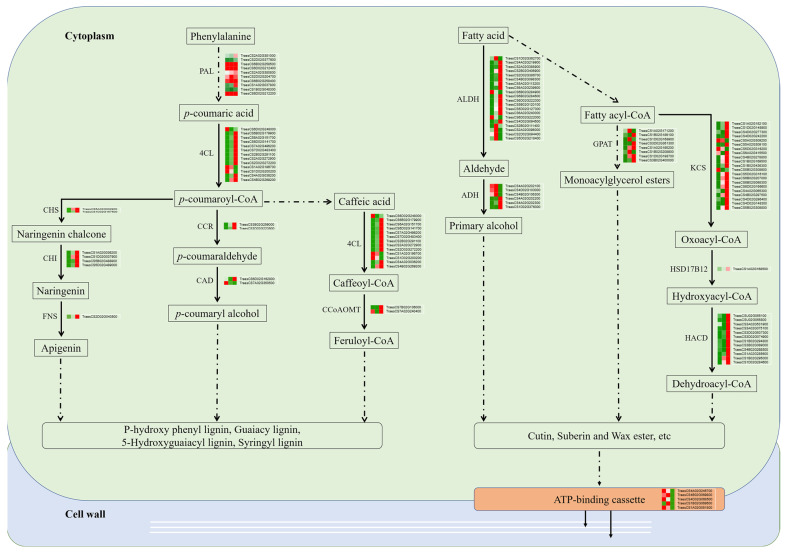
A simplified coordination process of cell wall organization during wheat spike development. Data were obtained using the log_2_FPKM of each gene. Red and green represent up- and downregulated genes, respectively. PAL(Phenylalanine ammonialyase); 4CL(4-coumarate:coenzyme A ligase); CCoAOMT(Caffeoyl-CoA 3-O-methyltransferase); CCR(Cinnamoyl-CoA reductase); CAD(Cinnamyl alcohol dehydrogenase); CHS(Chalcone synthase); CHI(Chalcone-flavonone isomerase); FNS(Flavone synthase); KCS(3-ketoacyl-CoA synthase); HSD17B12(Very-long-chain 3-oxoacyl-CoA reductase); HACD(Very-long-chain (3R)-3-hydroxyacyl-CoA dehydratase); ALDH(Aldehyde dehydrogenase); ADH(Alcohol dehydrogenase); GPAT(Glycerol-3-phosphate acyltransferase).

## Data Availability

We have archived the raw sequence data mentioned in this paper in the Genome Sequence Archive (GSA) at the National Genomics Data Center, China National Center for Bioinformation/Beijing Institute of Genomics, Chinese Academy of Sciences. The accession number for the data is GSA: PRJCA009595, and it can be accessed publicly at https://ngdc.cncb.ac.cn/gsa (accessed on 5 January 2023).

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
