# Peer review of "Transcriptome Dynamics during Spike Differentiation of Wheat Reveal Amazing Changes in Cell Wall Metabolic Regulators"

_ijms, 2023, doi:10.3390/ijms241411666_

Round 1

Reviewer 1 Report

In work, there is an unclear connection between spike differentiation and cell wall metabolic regulators. Please better explain it in the Introduction and the Discussion sections.

Section 2.4.

Why was examined the expression of these 15 genes? They were not connected to the transcriptome profile in this work and were not described in the Introduction.

The discussion is poor. 

In that section, the author shows results for the phenylpropanoid pathway not presented in the Results section. On the other hand, the author did not discuss the results shown in the Results section and generally did not discuss transcriptome dynamics.

The author did not discuss the problem outlined in the title.

Generally, the methodological layout of this work is unclear and not explained in the earlier parts.

The manuscript should be written in a passive voice.

Some sentences are unclear and poorly constructed.

Author Response

Dear editors and reviewers:

Re: Manuscript   ID: ijms-2494808 and Title: Transcriptome Dynamics during Spike Differentiation of Wheat Reveal Amazing Changes in Cell Wall Metabolic Regulators

Thank you for your precious comments and advice. Those comments are all valuable and helpful for revising and improving our paper and the essential guiding significance of our research. We have studied the comments carefully and have made corrections which we hope meet with approval. Revised portions are marked in red on the paper. We would love to thank you for allowing us to resubmit a revised copy of the manuscript, and we highly appreciate your time and consideration.

Sincerely.

Junjie Han.

Q1: In work, there is an unclear connection between spike differentiation and cell wall metabolic regulators. Please better explain it in the Introduction and the Discussion sections.

Response: We are incredibly grateful to the reviewers for their meticulous reading of our paper and invaluable feedback on potential issues. We have genuinely reflected upon these matters and are committed to making improvements. In this study, we conducted transcriptome sequencing at three stages of spike development and observed significant activity in genes associated with cell wall metabolism. Through transgenic verification of the cell wall gene, we have demonstrated its ability to regulate wheat spike development. Establishing a direct link between spike differentiation and cell wall metabolic regulatory factors remains challenging, with limited research on this aspect. Our study aims to bridge this gap, and we hope our efforts will be acknowledged. Once again, we would like to express our heartfelt gratitude for the reviewers’ diligent work.

Q2: Why was examined the expression of these 15 genes? They were not connected to the transcriptome profile in this work and were not described in the Introduction.

Response: We apologize for any potential misunderstanding caused by our previous statement. To verify the correctness of the Transcriptome data, we randomly selected the screened differential genes and performed qRT-PCR verification. We have selected at least 60 genes for validation work. However, we are still willing to take the reviewers' opinions and have replaced the genes to better correspond to the article's content.

Q3: In that section, the author shows results for the phenylpropanoid pathway not presented in the Results section. On the other hand, the author did not discuss the results shown in the Results section and generally did not discuss transcriptome dynamics.

Response: Thanks sincerely for the reviewers' valuable suggestions! We will incorporate reviewers' concerns and make the necessary modifications to ensure a more accurate presentation of the research results. The discussion section has undergone significant revisions to articulate our conclusions better. We hope these modifications meet reviewers' expectations and enhance the quality and credibility of the paper. We will sincerely consider every suggestion provided.

Q4: The author did not discuss the problem outlined in the title.

Response: We sincerely thank the reviewers for their careful reading and excellent suggestions on our manuscript. These suggestions are beneficial for further improving our research. We have noticed relevant issues and made significant revisions in the revised manuscript. The "Introduction" and "Discussion" sections reflect our main contributions. We hope this revision can improve the quality and readability of our paper. Once again, we express our gratitude for the reviewers' corrections and reviewers' patience. Reviewers' feedback is highly valued, and we are committed to continuous improvement in our research endeavours.

Q5: Some sentences are unclear and poorly constructed.

Response: We apologize for the language issues present in the original manuscript. To address this, we sought assistance from a native English speaker with a relevant research background to improve the language presentation.

Furthermore, we have modified the original manuscript to enhance its readability and better align it with the journal's requirements. We have already uploaded the revised manuscript and await approval from the reviewers. Once again, we are grateful for the valuable comments provided by the reviewers. Thank you!

Reviewer 2 Report

The manuscript entitled “Transcriptome dynamics during spike differentiation of wheat reveal amazing changes in cell wall metabolic regulators” by Han et. al. aimed to study the molecular mechanisms underlying the dynamic regulation of genes in wheats during spike differentiation. The authors investigated the gene expression at three different growth stages and found that high expression levels of genes relevant to the cell wall and carbohydrate metabolism. Further, they validated essential genes through transgenic experiments, which provide insights into the mechanism of wheat spike differentiation. 

Overall, this study is very interesting. The method used in the study is thorough. Conclusions are appropriate, and supported by the data. Statistical analysis is provided within the manuscript. Although the whole study is sound, there are minor concerns before I recommend accepting it. 

Minor concerns:

1. Line 94: “In comparing S2-vs-S1 and S3-vs-S2, we identified … ” should be “ We conducted a comparison between S2 and S1, and between S3 and S2, then identified many DEGs.”

2. Line 112-113: The authors mentioned “ The number of DEGs between S3 and S2 is twice that between S2 and S1, indicating that the latter period may hold more importance in biosynthesis and energy metabolism.” Does that mean most DEGs belong to the biosynthesis and energy metabolism pathway? Please clarify. 

3. It seems that the authors didn’t mention the accession number of transcriptomes data used in the manuscript. Please describe the sequencing data availability, so the readers have access to the data. 

4. Line 127: in the figure 2B legend, the author should explain the meaning of the numbers in the parenthesis nearby profile #4 (0.0, 1.0, 2.0) and profile #0 (0.0, -1.0, -2.0). 

5. Line 166: there is a missing space for “profile6”. 

6. It is recommended that the author should use the consistent figure panel alphabets. In figure 1, the authors used figure (a), (b). The rest of the figures panels were A, B, …. Please keep them consistent in the paper. 

7. Line 258: gene names should be italic. 

Minor editing of language

Author Response

Dear editors and reviewers:

Re: Manuscript   ID: ijms-2494808 and Title: Transcriptome Dynamics during Spike Differentiation of Wheat Reveal Amazing Changes in Cell Wall Metabolic Regulators

Thank you for your precious comments and advice. Those comments are all valuable and helpful for revising and improving our paper and the essential guiding significance of our research. We have studied the comments carefully and have made corrections which we hope meet with approval. Revised portions are marked in red on the paper. We would love to thank you for allowing us to resubmit a revised copy of the manuscript, and we highly appreciate your time and consideration.

Sincerely.

Junjie Han.

Q1: Overall, this study is very interesting. The method used in the study is thorough. Conclusions are appropriate, and supported by the data. Statistical analysis is provided within the manuscript. Although the whole study is sound, there are minor concerns before I recommend accepting it. 

Response: Thanks very much for taking the time to review this manuscript. We appreciate the reviewer’s positive evaluation of our work.

Q2: Line 94: “In comparing S2-vs-S1 and S3-vs-S2, we identified … ” should be “ We conducted a comparison between S2 and S1, and between S3 and S2, then identified many DEGs.”

Response: Thanks for the reviewer's careful review. We have made changes in the text based on the reviewer's comments to make our manuscript more readable.

Q3: Line 112-113: The authors mentioned “ The number of DEGs between S3 and S2 is twice that between S2 and S1, indicating that the latter period may hold more importance in biosynthesis and energy metabolism.” Does that mean most DEGs belong to the biosynthesis and energy metabolism pathway? Please clarify. 

Response: We apologize for any potential misunderstanding caused by our previous statement. Based on the differential genes we identified, the number of DEGs between S3-S2 is twice that of S2-S1. The development of wheat spikes involves many biosynthesis and energy metabolism processes, supporting spike differentiation and elongation, which is also demonstrated in GO enrichment. Therefore, many differentially expressed genes in the late stage of ear development are crucial for biosynthesis and energy transfer.

Q4: It seems that the authors didn’t mention the accession number of transcriptomes data used in the manuscript. Please describe the sequencing data availability, so the readers have access to the data. 

Response: We sincerely apologize for any inconvenience caused by the difficulty in discovering your login number. We mentioned the relevant letter in the final section of the manuscript, 'Data Availability Statement'.

Q5: Line 127: in the figure 2B legend, the author should explain the meaning of the numbers in the parenthesis nearby profile #4 (0.0, 1.0, 2.0) and profile #0 (0.0, -1.0, -2.0). 

Response: We profoundly apologize for any inconvenience caused by incomplete image information. The numbers near profile # 4 (0.0, 1.0, 2.0) and profile # 0 (0.0, -1.0, -2.0) represent the trend model in the corresponding Run chart. We have provided a more detailed description in the image description.

Q6: Line 166: there is a missing space for “profile6”. 

Response: Thank you for your careful review. We have made changes to the manuscript accordingly.

Q7: It is recommended that the author should use the consistent figure panel alphabets. In figure 1, the authors used figure (a), (b). The rest of the figures panels were A, B, …. Please keep them consistent in the paper. 

Response: We sincerely apologize for the issues that arose in the images. In response to the reviewer's feedback, we have revised the manuscript. We want to extend our heartfelt gratitude to the reviewers for their attentiveness and patience.

Q8: Line 258: gene names should be italic.

Response: We have reviewed our manuscript again and made the necessary corrections to the genes mentioned in the text. We greatly appreciate any feedback from the reviewer, as it is precious in improving our article.

Q9: Minor editing of language

Response: We apologize for the language problems in the original manuscript. The language presentation was improved with assistance from a native English speaker with an appropriate research background.

Furthermore, we have modified the original manuscript to enhance its readability and better align it with the journal's requirements. We have already uploaded the revised manuscript and await approval from the reviewers. Once again, we are grateful for the valuable comments provided by the reviewers. Thank you!

Round 2

Reviewer 1 Report

Dear Editor,

The manuscript has been corrected and in my opinion can be accepted in the present form.

Best Regards